# FumDSB Can Reduce the Toxic Effects of Fumonisin B_1_ by Regulating Several Brain-Gut Peptides in Both the Hypothalamus and Jejunum of Growing Pigs

**DOI:** 10.3390/toxins13120874

**Published:** 2021-12-07

**Authors:** Quancheng Liu, Fuchang Li, Libo Huang, Wenjie Chen, Zhongyuan Li, Chunyang Wang

**Affiliations:** 1Shandong Provincial Key Laboratory of Animal Biotechnology and Disease Control and Prevention, College of Animal Science and Veterinary Medicine, Shandong Agricultural University, 61 Daizong Street, Taian 271018, China; 2019110441@sdau.edu.cn (Q.L.); chlf@sdau.edu.cn (F.L.); huanglibo@sdau.edu.cn (L.H.); 2020110512@sdau.edu.cn (W.C.); 2Key Laboratory of Industrial Fermentation Microbiology, College of Biotechnology, Tianjin University of Science and Technology, Tianjin 300457, China; 3Tianjin Key Laboratory of Industrial Microbiology, Tianjin University of Science and Technology, Tianjin 300457, China

**Keywords:** fumonisin B_1_, carboxylesterase FumDSB, hypothalamus, jejunum, brain-gut peptide, ghrelin, obestatin, growing pigs

## Abstract

Fumonisin B_1_ (FB_1_) is the most common food-borne mycotoxin produced by the *Fusarium* species, posing a potential threat to human and animal health. Pigs are more sensitive to FB_1_ ingested from feed compared to other farmed livestock. Enzymatic degradation is an ideal detoxification method that has attracted much attention. This study aimed to explore the functional characteristics of the carboxylesterase FumDSB in growing pigs from the perspective of brain–gut regulation. A total of 24 growing pigs were divided into three groups. The control group was fed a basal diet, the FB_1_ group was supplemented with FB_1_ at 5 mg/kg feed, and the FumDSB group received added FumDSB based on the diet of the FB_1_ group. After 35 days of animal trials, samples from the hypothalamus and jejunum were analyzed through HE staining, qRT-PCR and immunohistochemistry. The results demonstrated that the ingestion of FB_1_ can reduce the feed intake and weight gain of growing pigs, indicating that several appetite-related brain-gut peptides (including NPY, PYY, ghrelin and obestatin, etc.) play important roles in the anorexia response induced by FB_1_. After adding FumDSB as detoxifying enzymes, however, the anorexia effects of FB_1_ were alleviated, and the expression and distribution of the corresponding brain-gut peptides exhibited a certain degree of regulation. In conclusion, the addition of FumDSB can reduce the anorexia effects of FB_1_ by regulating several brain-gut peptides in both the hypothalamus and the jejunum of growing pigs.

## 1. Introduction

As one of the main food-borne mycotoxins, fumonisins are produced by *F. verticillioides*, *F. moniliforme*, *F. proliferatum* and some related species, which are widely found in various grains around the world, especially in corn [1]. To date, fumonisin B_1_ (FB_1_) has the highest levels of contamination among all the 28 types of fumonisins isolated in nature, accounting for about 70–80% of naturally contaminated food and feed [2]. As a consequence, toxicological studies concerning fumonisins have focused on FB_1_. Due to the stability of chemical structures at high temperatures, FB_1_ is not easily removed during food or feed processing; accordingly, it causes acute or chronic poisoning in humans and animals through the food chain [3,4,5]. As a typical monogastric livestock, the gastrointestinal tract of pigs lacks sufficient microbiota to degrade or destroy mycotoxins [6]. Due to the higher proportion of corn in the complete feed of growing and fattening pigs, they are more sensitive to fumonisins ingested from feed in comparison to other farmed livestock, such as cattle and chickens [7,8].

Although many efforts have been carried out, food safety problems induced by mycotoxins are still inevitable. Some scholars even pointed out that the health problems of different pig breeds in pig farms are often directly or indirectly caused by the presence of mycotoxins in full-price feed [9]. Therefore, efficient and safe detoxification methods have attracted wide attention from breeding enterprises and veterinary experts. Most physical and chemical detoxification methods easily destroy the nutrient composition of feed and change its palatability, thereby affecting the quality and yield of animal products [4,10]. The biological detoxification method is used to convert mycotoxins into low-toxic or non-toxic degradation products through the use of live microorganisms or biological enzymes, which are widely favored due to their high degradation efficiency and relative safety [11]. FUMzymes, developed by Biomin Co., acting as biodegradable enzymes on FB_1_, were certified by the European Union in 2014 [12]. Since then, increasing numbers of live bacteria or biological enzymes designed to detoxify FB_1_ have been screened and identified [13,14,15]; however there are fewer studies on the detoxification mechanism in vivo of biological enzymes. FumDSB is a new type of carboxylesterase, which can degrade FB_1_ to hydrolyzed fumonisin B_1_ (HFB_1_) by releasing two tricarballylic acid groups, thereby reducing the toxicity of FB_1_ [16]. Purified FumDSB showed the strongest ability to destroy FB_1_ at pH 6.0, and exhibited favorable stability in a wide range of pH, from 4.0 to 9.0. In addition, the optimum operating temperature of FumDSB was 30 °C, and the relative activities of 67.33% and 34.73% were maintained at 37 °C and 40 °C, respectively [16]. In view of its excellent pH stability and moderate thermal stability, FumDSB is very suitable for use under animal physiological conditions. However, it is necessary to further analyze its effects and safety in vivo.

Many studies have shown that the brain–gut axis, acting as a complex neuro-endocrine network between the brain and the gastrointestinal tract, is closely related to appetite regulation [17]. As the material basis of the brain–gut axis, brain-gut peptides are dually distributed in the brain and the gastrointestinal tract. On the one hand, brain-gut peptides act as hormones to mediate brain–gut regulation [18]. On the other hand, they interact with the sensory nerve endings or corresponding receptors of smooth muscle cells of the gastrointestinal tract, thereby playing a role in the regulation of gastrointestinal function [19]. In addition to its effects on the gut, FB_1_ is also a neurotoxin that can cause damage to the nerve cell [20]. Gbore found that the acetylcholinesterase activity in the local brain and pituitary changed significantly after feeding weaned piglets with a diet containing 5 mg/kg FB_1_ [21]. Dietary exposure to FB_1_ at an approximate concentration of 5 mg/kg FB_1_ or higher is a potential health risk for growing pigs, which may lead to neurochemical changes in the brain, and even induce adverse physiological reactions [22]. After the pigs were supplied with the grains contaminated by deoxynivalenol (DON) and fumonisin, the concentration of 5-hydroxytryptamine (5-HT) increased linearly in the brain, and the peripheral adrenergic system was damaged, thereby affecting endocrine regulation [23]. 

These findings indicated that brain–gut regulation may play an important role in the reduction in feed intake and weight loss of animals induced by FB_1_. The hypothalamus and the jejunum are the two main organs related to brain–gut regulation. Therefore, the purpose of this study is to evaluate the effects of carboxylesterase FumDSB on appetite-related brain-gut peptides in the hypothalamus and jejunum of growing pigs, and then to explore the mechanism and safety of biodegradable enzymes from the perspective of brain–gut regulation. 

## 2. Results

### 2.1. Production Performance

The effects of the addition of FumDSB and FB_1_ on the production performance of growing pigs are listed in Table 1. Compared to the control group, the average daily feed issue (ADFI) and average daily body weight gain (ADG) decreased significantly (*p* < 0.05), both in the FB_1_ group and FumDSB group, while the feed ratio (F/G) increased significantly (*p* < 0.05). Meanwhile, ADFI and ADG increased significantly more in the FumDSB group than in the FB_1_ group, while F/G decreased significantly (*p* < 0.05). 

### 2.2. mRNA and Protein Expression of Several Brain-Gut Peptides in Hypothalamus and Jejunum

The effects of the addition of FB_1_ and FumDSB on the mRNA expression of several brain-gut peptides are displayed in Figure 1. Compared with the control group, the mRNA expression of Neuropeptide Y (NPY) in the hypothalamus decreased significantly (*p* < 0.05) in the FB_1_ group, while that of Peptide YY (PYY) and the receptor 2A of 5-HT (5-HT2A) increased significantly (*p* < 0.05). In the FumDSB group, the mRNA expression of NPY in the hypothalamus increased significantly (*p* < 0.05) compared with the FB_1_ group, while that of PYY and 5-HT2A decreased significantly (*p* < 0.05) (Figure 1a). Meanwhile, the data from the jejunum demonstrated that the mRNA expression of NPY decreased significantly (*p* < 0.05) in the FB_1_ group compared with the control group, while that of 5-HT2A increased significantly (*p* < 0.05). Compared with the FB_1_ group, the mRNA expression of NPY increased significantly (*p* < 0.05) in the FumDSB group, while that of 5-HT2A decreased significantly (*p* < 0.05). However, there were no significant differences (*p* > 0.05) for the mRNA expression of PYY in the jejunum among the three groups (Figure 1b).

The results from the western blot (Figure 2) revealed that the relative protein expression levels of NPY in the hypothalamus decreased significantly (*p* < 0.05) compared with the control groups, while that of PYY increased significantly (*p* < 0.05). In the FumDSB group, the relative protein expression levels of NPY increased significantly (*p* < 0.05) compared with the FB_1_ group, while that of PYY decreased significantly (*p* < 0.05). There were no significant differences (*p* > 0.05) for the protein expression of 5-HT2C in the jejunum among the three groups (Figure 2a). Meanwhile, the data from the jejunum demonstrated that the protein expression of NPY decreased significantly (*p* < 0.05) in the FB_1_ group compared with the control group, while that of PYY and 5-HT2C increased significantly (*p* < 0.05). Compared with the FB_1_ group, the protein expression of PYY and 5-HT2C decreased significantly (*p* < 0.05) in the FumDSB group, but that of PYY exhibited no significant differences (*p* > 0.05) (Figure 2b).

### 2.3. Distribution of Several Brain-Gut Peptides in the Hypothalamus

The effects of the addition of FB_1_ and FumDSB on the distribution of several brain-gut peptides in the hypothalamus are illustrated in Figure 3. Typical pictures of immunohistochemistry (IHC) demonstrated that the positive reaction substances of NPY were diffuse, and mainly concentrated in the cytoplasm of nerve cells, or near the hypothalamic nerve cells. Moreover, the positive reaction both in the FB_1_ group and the FumDSB group was weaker than that of the control group (Figure 3a). The immunopositive spots of PYY mainly gathered in the cytoplasm and cell membrane of neuronal cells. Meanwhile, the immunopositive reactions of PYY were stronger than in the control group after adding FB_1_. The immunopositive reaction in the FB_1_ group was stronger than that of the control group and the FumDSB group, but the degree of positive reaction in the FumDSB group was similar to that in the control group (Figure 3b). Yellow-brown positive reaction spots of 5-HT2C took the form of round pellets, which were filled in the neuronal cells. The positive reactions of 5-HT2C in the FB_1_ group were stronger than in the control group and the FumDSB group (Figure 3c). 

### 2.4. Distribution of Several Brain-Gut Peptides in the Jejunum

The effects of the addition of FB_1_ and FumDSB on the distribution of several brain-gut peptides in the jejunum are illustrated in Figure 4. The results of the IHC demonstrated that NPY, PYY, and 5-HT2C were all distributed in the jejunum. The diffuse positive reaction spots of NPY were mainly located at the top of the jejunal villi, and were partly found near the small intestinal glands. The positive reactions of the FB_1_ group and the FumDSB group were significantly weaker compared with the control group, but the reaction degree in the FB_1_ group was stronger than those in the FumDSB group (Figure 4a). The positive reaction spots of PYY were mainly concentrated in the villous epithelium of the jejunum, in the villous lamina propria and in the vicinity of the small intestinal glands. The positive reactions in the FB_1_ group and the FumDSB group were significantly stronger than those in the control group (Figure 4b). The positive reaction spots of 5-HT2C positive were mainly scattered in the lamina propria of the jejunum, as well as in the glands of the small intestine. The immunopositive spots in the FB_1_ group and the FumDSB group were significantly stronger than those in the control group (Figure 4c).

### 2.5. mRNA and Protein Expression of Ghrelin/Obestatin Receptor in Hypothalamus and Jejunum

The effects of the addition of FB_1_ and FumDSB on the mRNA expression levels of ghrelin/obestatin preprohormone (GHRL), growth hormone secretagogue receptor (GHSR) and G protein-coupled receptor 39 (GPR-39) are displayed in Figure 5. The results demonstrated that in the hypothalamus, the mRNA expression levels of GHRL and GPR-39 increased significantly in the FB_1_ group (*p* < 0.05) compared with the control group, while that of GHSR decreased significantly (*p* < 0.05). In the FumDSB group, the mRNA expression of GHRL and GPR-39 decreased significantly (*p* < 0.05) compared with the FB_1_ group, but there were no significant differences (*p* > 0.05) in the mRNA expression of GHSR (Figure 5a). Meanwhile, the data from the jejunum showed that the mRNA expression of GHRL and GPR-39 increased significantly (*p* < 0.05) in the FB_1_ group compared with the control group. Compared with the FB_1_ group, the mRNA expression of GHRL decreased significantly (*p* < 0.05) in the FumDSB group. However, there were no significant differences (*p* < 0.05) for the mRNA expression of GHSR in the jejunum among the three groups (Figure 5b).

The data from the western blot (Figure 6) showed that the relative protein expression levels of obestatin in the hypothalamus increased significantly (*p* < 0.05) compared with the control groups, while that of obestatin decreased significantly (*p* < 0.05) in the FumDSB group compared with the FB_1_ group. There were no significant differences (*p* > 0.05) for the protein expression of ghrelin and GHSR among the three groups (Figure 6a). Meanwhile, the data from the jejunum demonstrated that the protein expression of Ghrelin and GHSR decreased significantly (*p* < 0.05) in the FB_1_ group compared with the control group, while that of obestatin increased significantly (*p* < 0.05). In the FumDSB group, the protein expression of GHSR increased significantly compared with the FB_1_ group, while that of obestatin decreased significantly (*p* < 0.05) (Figure 6b).

### 2.6. Distribution of Ghrelin/Obestatin/GHSR in Hypothalamus

The typical IHC pictures showed that the immunopositive reaction substances of ghrelin were mainly distributed near the neuroendocrine cells, and the positive reaction spots of ghrelin in the FB_1_ group were significantly lower than those in the control group and the FumDSB group (Figure 7a). The obestatin immunopositive reaction substances were distributed near the nucleus of the hypothalamic nerve cells, and the positive reaction spots of obestatin in the FB_1_ group were significantly stronger than those in the control group and the FumDSB group (Figure 7b). The distribution of the positive GHSR reaction spots, which were mainly distributed around the nerve cells, were mainly diffuse and irregular in shape. Furthermore, the positive GHSR reaction spots in the FB_1_ group were slightly lower than those in the control group (Figure 7c).

### 2.7. Distribution of Ghrelin/Obestatin/GHSR in Jejunum

The IHC results demonstrated that the immunopositive reaction signals of ghrelin were mainly distributed in the small intestinal mucosal epithelium of the jejunum and at the top of the jejunal villi, near the intestinal lumen, but less in the small intestinal glands. The reaction degree in the control group was stronger than in the FB_1_ and FumDSB groups, while the reaction degree in the FumDSB group was slightly stronger than in the FB_1_ group (Figure 8a).The obestatin immunopositive reaction spots in the control group mainly scattered in the glandular epithelium of the small intestine, but rarely in the mucosal epithelium. However, there was a large number of strong positive reaction spots in the mucosal epithelium of the FB_1_ group. Furthermore, in the FumDSB group, positive signals were observed in the epithelial cells on both sides of the villi of the jejunum and also near the intestine glands. The reaction degree in the FB_1_ group and the FumDSB group was stronger than in the control group (Figure 8b). The immunopositive reaction spots of GHSR in the control group were mainly concentrated in the jejunal villi epithelium; in the FB_1_ group they were mainly distributed in the mucosal epithelial cells at the top of the villi; and in the FumDSB group, they could be observed not only in the small intestinal glandular epithelium, but also in the epithelial cells on both sides of the jejunal villi. The reaction intensity in the control group was significantly stronger than in the FB_1_ group and the FumDSB group (Figure 8c).

## 3. Discussion

### 3.1. Selection of Additive Amount of FB_1_


FB_1_ is one of the natural mycotoxins with the highest pollution rate in feed materials, especially in corn. Acute poisoning induced by FB_1_ at high concentrations is very rare in modern breeding environments. However, because growing pigs are more sensitive to FB_1_, some adverse toxic effects have been found in pigs after the ingestion of FB_1_ at medium and low doses [24,25]. In addition to classic pulmonary edema and injury to the liver and kidneys, the destruction of the digestive function induced by FB_1_ is also increasingly concerning. Results from in vitro and in vivo experiments proved that the ingestion of FB_1_ can damage the barrier function of intestinal epithelial cells, destroy tissue morphology, induce intestinal inflammatory response and increase the chances of infection, thereby affecting the nutrient absorption and growth of animals [8,26,27]. At present, except for poultry and ruminants, the maximum limit on the use of fumonisin in full-price feeds of horses, rabbits, pigs and other animals is usually at 5 mg/kg in many countries [28,29,30]. When the additional amount of FB_1_ in pig feed is 5 mg/kg or higher for 6 months, it is likely to cause neurochemical changes in the brain and lead to undesirable physiological reactions [22]. Therefore, the critical limit of 5 mg/kg was selected as the challenge dose to study the effects of FB_1_ on the hypothalamus and jejunum in this study. 

### 3.2. Appetite-Related Brain-Gut Peptide 

The results of this study demonstrated that no obvious clinical symptoms of poisoning, such as diarrhea, abdominal pain, or even death, were observed in growing pigs after ingesting FB_1_ but th pigs’ food intake and weight gain were significantly reduced. Studies suggested that abnormal secretion and unbalanced regulation of brain-gut peptides are one of the main reasons for food intake and energy metabolism [31]. According to their different effects on appetite regulation, brain-gut peptides can be divided into two categories, namely appetite-promoting (including NPY, ghrelin, etc.) and appetite-suppressing (including PYY, 5-HT, obestatin, etc.) [32,33]. In this study, the expression and distribution of NPY, PYY and the receptor of 5-HT in the hypothalamus and jejunum were detected. 

NPY, first isolated from pigs, is one of the most highly conserved neuropeptides in vertebrates. It is a single-chain polypeptide with 36 amino acid residues and a molecular weight of 4.2 KD [34]. As an important appetite-stimulating factor in the hypothalamic appetite regulation network, NPY can regulate the energy balance of the body and enhance the peristalsis of the duodenum, jejunum and colon, thereby regulating the feeding behavior of animals [35]. PYY, which is secreted by pancreatic endocrine cells (PP cells), is a short peptide composed of 36 amino acids and is mainly distributed in the nervous system and gastrointestinal tract [36]. Although PYY and NPY share a 70% homology, they exert opposite effects on appetite regulation [34]. Acting as a kind of brain-gut peptide providing satiety cues and anorexia signals, the main function of PYY is to reduce the peristalsis of the intestines and the secretion of digestive juices by inhibiting NPY neurons, thereby inhibiting food intake [37]. Serotonin, also known as 5-HT, is a biogenic amine and tryptophan derivative. An important inhibitory neurotransmitter, 5-HT participates in the regulation of various physiological activities in animals [38]. Only after binding with its corresponding receptor can 5-HT exert these physiological effects. The receptors of 5-HT can be divided into 14 subtypes, and the 5-HT2 receptor family, 5-HT2A and 5-HT2C are closely connected with the regulation of appetite and energy balance [39]. The 5-HT2A receptor and 5-HT2C receptor are similar in structure, pharmacological characteristics and cell signal transduction pathway [40]. Several studies have proposed that under specific stimulation, 5-HT can activate anorectic pro-opiomelanocortin (POMC) neuron and NPY simultaneously, thereby reducing food intake, promoting satiety and affecting energy metabolism [39,41]. The three appetite-related brain-gut peptides interact with each other to regulate feeding behavior jointly.

At present, research concerning the effects of brain-gut peptides on lower appetite induced by the consumption of FB_1_ is very limited. In this study, several appetite-related brain-gut peptides were dually distributed in the hypothalamus and jejunum. After adding FB_1_ to feed, the expression of NPY (appetite-promoting) in both the hypothalamus and the jejunum of growing pigs reduced significantly, while that of PYY and the receptor 5-HT (appetite-suppressing) increased significantly. These results indicated that as the material basis of brain–gut regulation, brain-gut peptides play an important role in appetite suppression induced by FB_1_. It was particularly noteworthy that the mRNA and protein expression of PYY in the hypothalamus was higher than in the jejunum, suggesting that PYY may be more sensitive to FB_1_. After adding FumDSB as a detoxification enzyme, the toxic effect of FB_1_ on the feed intake of growing pigs was alleviated significantly, and the expression and distribution of the NPY, PYY and 5-HT receptors exhibited a certain degree of regulation.

### 3.3. Ghrelin/Obestatin in the Hypothalamus and Jejunum Induced by FB_1_

Ghrelin and obestatin are considered important brain-gut peptides in the current research on the brain–gut axis. As a small molecule polypeptide containing 28 amino acid residues, ghrelin is an endogenous ligand of GHSR; it is mainly expressed in the hypothalamus, pituitary, vagus afferent nerve fibers and gastrointestinal myenteric plexus [42]. After binding to its receptor, ghrelin can stimulate the release of growth hormone (GH), promote appetite and fat deposition, increase body weight, enhance gastrointestinal motility and participate in regulating energy balance, as well as other physiological effects [43]. Obestatin is a small peptide consisting of 23 amino acid residues, derived from the same precursor gene as ghrelin, GHRL. After binding to its receptor, GPR-39, obestatin can antagonize the effects of ghrelin, including the suppression of appetite, delaying gastric emptying, inhibiting jejunal movement and reducing weight [44]. Ghrelin, obestatin and their receptors interact with the central appetite regulation network through the brain–gut axis, which can regulate food or feed intake, affect gastrointestinal function, control body weight and play an important role in the regulation of animal feeding. The dynamic balance of the two functions may be an important mechanism for the body to regulate energy metabolism and body weight balance [45]. In this study, the mRNA expression of GHRL and its receptors in the hypothalamus and jejunum were detected, and the results demonstrated that the mRNA expression of GHSR both in the hypothalamus and the jejunum decreased significantly, while that of GHRL and GPR-39 only increased significantly in the jejunum. The results from western blot and the IHC highlighted that the expression and distribution of ghrelin decreased in the hypothalamus and jejunum, while that of obestatin increased, indicating that ghrelin and obestatin play an important role in the anorexia response of growing pigs induced by FB_1_. After adding FumDSB as a detoxifying enzyme, the expression and distribution of ghrelin and obestatin exhibited a certain degree of regulation. 

The results of LC-QTOF-MS in a previous study proved that FumDSB can degrade FB_1_ into hydrolyzed fumonisin B_1_ (HFB_1_) by releasing two tricarballylic acid groups. Furthermore, after removing these two TCA side chains, HFB_1_ is a ten times weaker inhibitor of ceramide synthase than FB_1_, and its cytotoxicity is also reduced [27,46]. In the present study, the results demonstrate that FumDSB can reduce the negative effects of FB_1_ on the production performance of growing pigs. This may be because FumDSB can degrade FB_1_ into HB_1_, thereby reducing the toxic effects of FB_1_. In short, FumDSB can be used as a potential food and feed additive to reduce the toxicity of FB_1_.

## 4. Conclusions

In conclusion, FB_1_ ingestion can reduce the expression of appetite-promoting factors and increase the expression of appetite-suppressing factors. However, FumDSB can reduce the toxic effects of FB_1_ by regulating several brain-gut peptides in both the hypothalamus and the jejunum of growing pigs.

## 5. Materials and Methods

### 5.1. Ethics Statement 

All the experimental schemes concerning living animals complied with ethical regulations, which were approved by the Animal Protection Committee of Shandong Agricultural University (ACA-2020-056, Taian, Shandong, China).

### 5.2. Source and Pre-Processing of Fumonisin Detoxification Enzymes, FumDSB 

FumDSB (Patent No: ZL201811171697.3) is a new type of carboxylesterase derived from *Sphingomonas bacteria* (KUO56785), expressed in *Escherichia coli*. It was provided by Dr. Zhongyuan Li of Tianjin University of Science and Technology. The primary experiment proved that FumDSB can degrade FB_1_ into hydrolyzed fumonisin B_1_ (HFB_1_) by releasing two tricarballylic acid groups, and that the degradation rate can reach 100% [16]. 

### 5.3. Animal Feeding and Sampling

In total, 1 g of powdered FB_1_ standard with a chromatographic purity of 98% (Triplebond Company, Guelph, ON, Canada) was dissolved in 1 L of ethyl acetate. After spraying using this mixed solution, 1 kg talc carrier was placed overnight so as to volatilize the ethyl acetate ester. Noxious corn flour containing 10 mg/kg FB_1_ was prepared by mixing the talc carriers containing FB_1_ into corn flour in order to replace the corn flour used in the feed formula for this experiment. 

The composition and nutrients of the basal diet used in this study, which was formulated based on the NRC (1998) recommendations, are presented in Table 2. The basal diet without any fungi-remove agent was prepared before the formal experiment and stored in a ventilated and dry place. The feed samples were collected according to the standard method. Subsequently, the content of FB_1_ + FB_2_, zearalenone (ZEA), deoxynivalenol (DON) and aflatoxin B_1_ (AFB_1_) was detected in Romer Laboratory (Wuxi, Jiangsu, China). The results are listed in Table 2. 

Twenty-four two-month-old, healthy, three-way crossbred growing pigs (Duroc × Landrace × Large white) were randomly divided into three treatment groups based on their initial weight, with the same number of males and females, with eight pigs and four replicates per treatment. The control group included the pigs on the basal diet containing normal corn flour, the FB_1_ group was fed with the basal diet containing noxious corn flour, and the final content of FB_1_ in the feed was at 5 mg/kg. In addition, the diet of the FumDSB group was mixed with 0.1% FumDSB as an enzyme preparation on the basis of the FB_1_ group feed, and the FumDSB added and mixed once a week. During the trial period, water and feed were supplied ad libitum, and the feed residue in the trough was collected and weighed every day. After the feeding trial, all the pigs were weighed individually, and then the values of ADG, ADFI and F/G were calculated according to the conventional test methods. The feeding trial was carried out on Baoding Farm in Hebei Province in China. Meanwhile, the immunization procedures and management standards were also executed in accordance with routine procedures. The complete animal feeding process included a 7d adaptation period and a 35d formal trial period. 

At the end of the feeding trials, six pigs from each treatment group with an empty stomach were selected and euthanized by electric shock. The samples of the hypothalamus and jejunum located in the same region were collected accurately after the experimental pigs were quickly dissected. After washing with physiological saline, part of the samples was fixed using a 4% paraformaldehyde solution (P395744, ALADDIN Co., Shanghai, China), while the rest were stored at −80 °C for later use.

### 5.4. qRT-PCR

The total RNA of the hypothalamus and jejunum were extracted using a Trizol RNA Extraction Kit (15596026, Thermo Fisher, Waltham, MA, USA), and the integrity of the RNA was verified through 0.8% agarose electrophoresis. After the quality and concentration of the RNA were detected using NanoDrop 2000 ultra-micro UV spectrophotometer (Thermo Fisher, MA, USA), the total RNA was transcribed into cDNA using an EasyScript cDNA Synthesis SuperMix kit (DP419, Transgen Co., Beijing, China). Next, the cDNA was mixed with SYBR Green Premix Pro Taq HS qPCR Kit (AG11718, Accurate Bio., Changsha, Hunan, China) and applied to Applied Biosystems 7500 (Thermo Fisher, MA, USA). Several target genes were selected in this experiment, including NPY, PYY, 5-HT2A, GHRL, GHSR and GPR39, while glyceraldehyde-3-phosphate dehydrogenase (GAPDH) and β-actin served as double internal references. All the primer sequences used in this study, which were supplied by Shanghai Sangon Biotech, are listed in Table 3. Each qRT-PCR reaction system contained 2 μL templates, 0.4 μL primers and 10 μL SYBR green master mixes. After the circulation index (Ct) of the samples was measured, the relative expression of the genes was calculated using the 2^−ΔΔCt^ method [40]. The double internal parameter method was used in this study, and all target gene products were specifically verified.

### 5.5. Western Blot

After being homogenized at a low temperature through the addition of nine times the volume of RIPA (P0013b, Beyotime Bio., Beijing, China) and the protease inhibitor (P1005, Beyotime Bio., Beijing, China), the tissue samples from the hypothalamus and jejunum were crushed using ultrasound and centrifuged at 12,000 rpm for 10 min. After the supernatants were collected, a BCA kit (P0012, Beyotime Bio., Beijing, China) was used to quantify the protein. The SDS-PAGE (PG112, Epizyme Biotech, Shanghai, China) was used to separate the proteins, and a PVDF membrane (IPVH00010, Millipore, Darmstadt, Germany) was used for the membrane transfer experiment. In this study, the PVDF membranes were incubated with primary antibodies overnight at 4 °C, including against polyclonal rabbit antibody NPY, PYY, 5-HT2C, ghrelin, obestatin and GHSR (1:1000, BIOSS, Beijing, China). Subsequently, the goat anti-rabbit IgG, as the secondary antibody, was labeled using HRP (1:2000, A0208, Beyotime Bio., Beijing, China). Finally, a Fusion imaging system (Fusion FX, Vilber, Paris, France) was used to take the photographs and the optical density were calculated using Image-Pro Plus 6.0 (Image Pro-Plus 6.0, Media Cybernetics, Silver Spring, MD, USA).

### 5.6. Immunohistochemistry (IHC) 

The tissue samples fixed with paraformaldehyde were dehydrated and wrapped in paraffin blocks. Next, they were cut into 5 μm sections and fixed on Poly-L-Lysine slides. These tissue sections were subjected to antigen thermal repair after xylene dewaxing and alcohol dehydration. A variety of primary polyclonal rabbit antibodies was used after serum blockage in this study, including NPY, PYY, 5-HT2C, ghrelin, obestatin and GHSR antibodies (1:150, BIOSS, Beijing, China). Next, the secondary antibody was a rabbit two-step assay kit (PV-9001, ZSBIO, Beijing, China), whcih used for subsequent treatment [47]. After DAB (PA110, Transgen Co., Beijing, China) staining and a series of alcohol dehydration steps were carried out, the tissue sections were sealed using neutral resin and cover glass. Finally, the slices were observed using a microscope (Nikon Co., Tokyo, Japan).

### 5.7. Data Statistics 

All experimental data results were calculated using excel. And all statistical analysis data were performed using SPSS 22 software (IBM Co., New York, NY, USA). In this experiment, a one-way ANOVA test was used and the LSD test was selected.The results of the experimental data are expressed as the mean ± standard deviation, and *p* < 0.05 was considered to indicate statistically significant differences. GraphPad Prism 8 was used to draw data images.

## Figures and Tables

**Figure 1 toxins-13-00874-f001:**
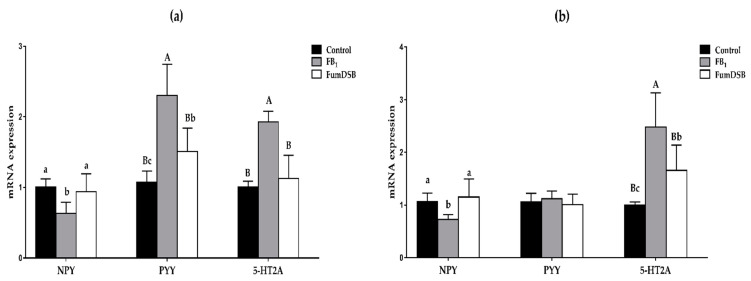
The mRNA expression of NPY\PYY\5-HT2A in the hypothalamus (**a**) and jejunum (**b**). Control, FB_1_ and FumDSB mean different treatment. Data are expressed as means ± standard deviations, *n* = 6. Different lowercase letters and capital letters indicate the significant difference at *p* < 0.05 and *p* < 0.01, respectively.

**Figure 2 toxins-13-00874-f002:**
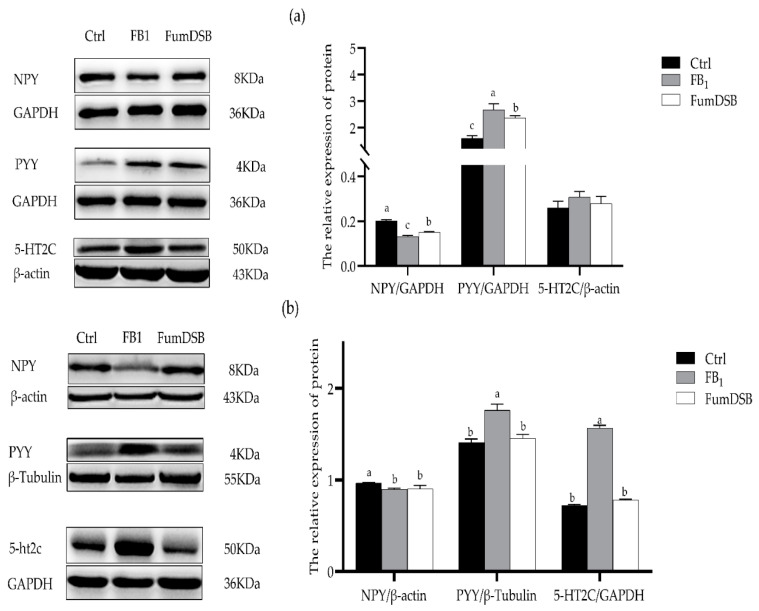
Relative protein expression of NPY\PYY\5-HT2C in the hypothalamus (**a**) and jejunum (**b**). Control, FB_1_ and FumDSB mean different treatment. Data are expressed as means ± standard deviations, *n* = 6. Different lowercase letters indicate the significant difference at *p* < 0.05.

**Figure 3 toxins-13-00874-f003:**
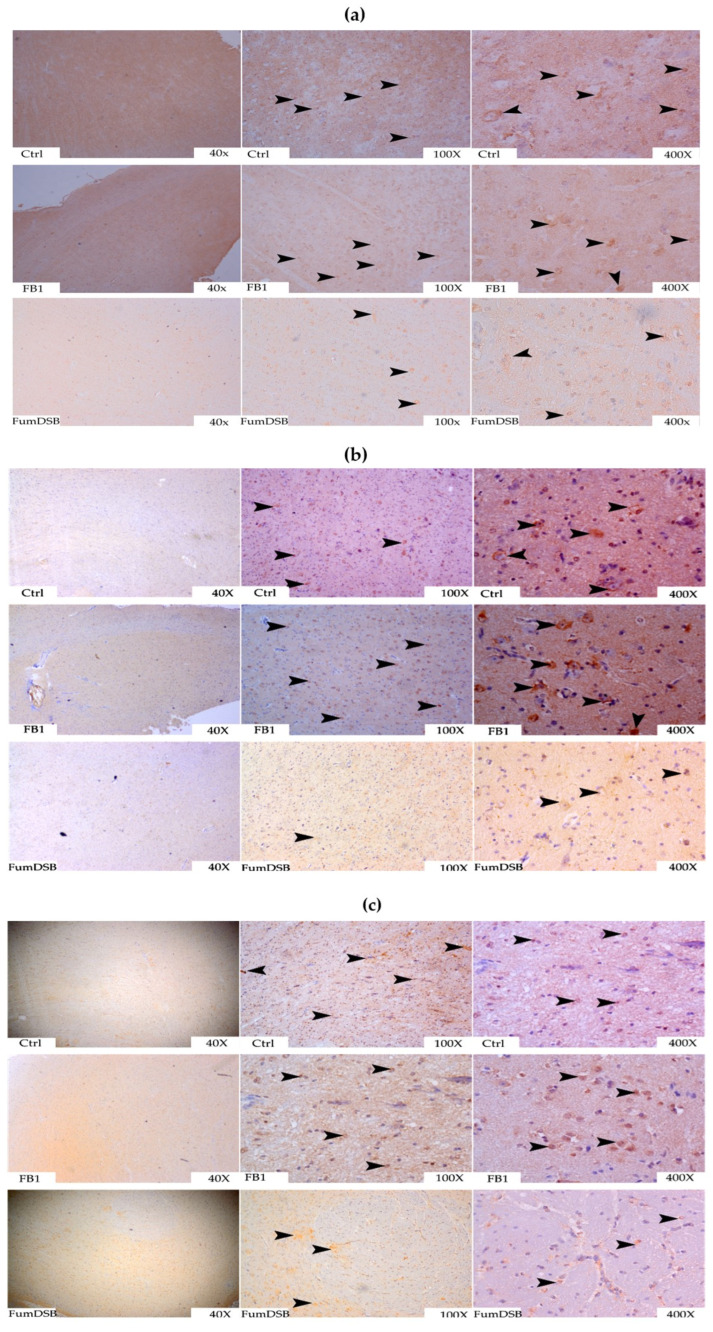
Representative IHC photograph of appetite-related brain-gut peptide in the hypothalamus. (**a**–**c**) means the results of NPY, PYY and 5-HT2C, respectively. Ctrl, FB,1 and FumDSB refer to different treatment. 40×, 100× and 400× represent the magnification. Brown positive reactants are emphasized using black arrows.

**Figure 4 toxins-13-00874-f004:**
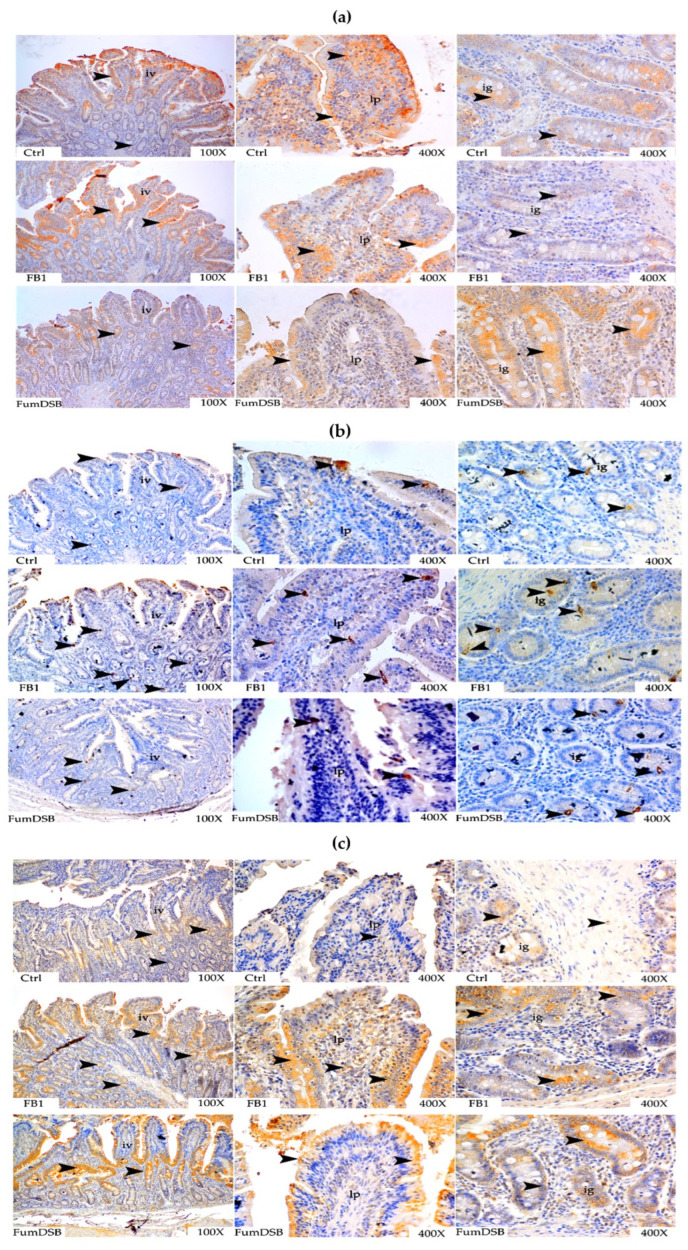
Representative IHC photograph of appetite-related brain-gut peptide in the jejunum. (**a**–**c**) means the results of NPY, PYY and 5-HT2C, respectively. Ctrl, FB_1_ and FumDSB refer to different treatment. iv means intestinal villus, ie means intestinal epithelium and ig means intestinal gland. 100× and 400× represent the magnification. Brown positive reactants are emphasized using black arrows.

**Figure 5 toxins-13-00874-f005:**
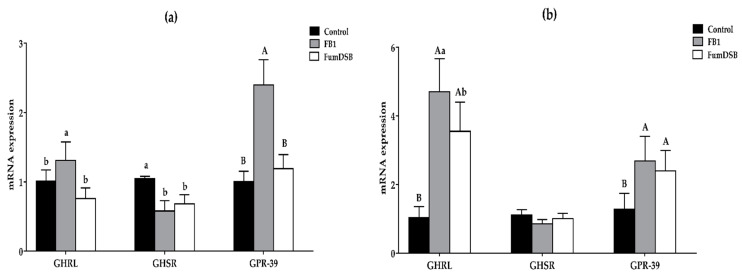
Relative mRNA expression of receptor of ghrelin/obestatin in the hypothalamus (**a**) and jejunum (**b**). Control, FB_1_ and FumDSB mean different treatment. Data are expressed as means ± standard deviations, *n* = 6. Different lowercase letters and capital letters indicate the significant difference at *p* < 0.05 and *p* < 0.01, respectively.

**Figure 6 toxins-13-00874-f006:**
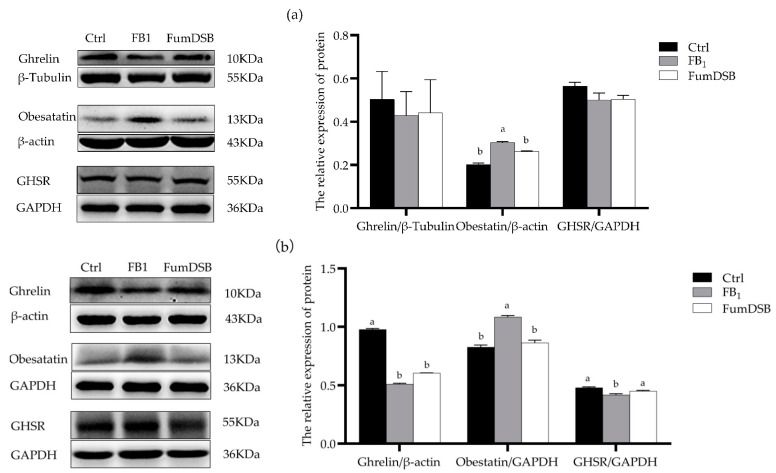
Relative protein expression of ghrelin/obestatin/GHSR in the hypothalamus (**a**) and jejunum (**b**). Control, FB_1_ and FumDSB mean different treatment. Data are expressed as means ± standard deviations, *n* = 6. Different lowercase letters indicate the significant difference at *p* < 0.05.

**Figure 7 toxins-13-00874-f007:**
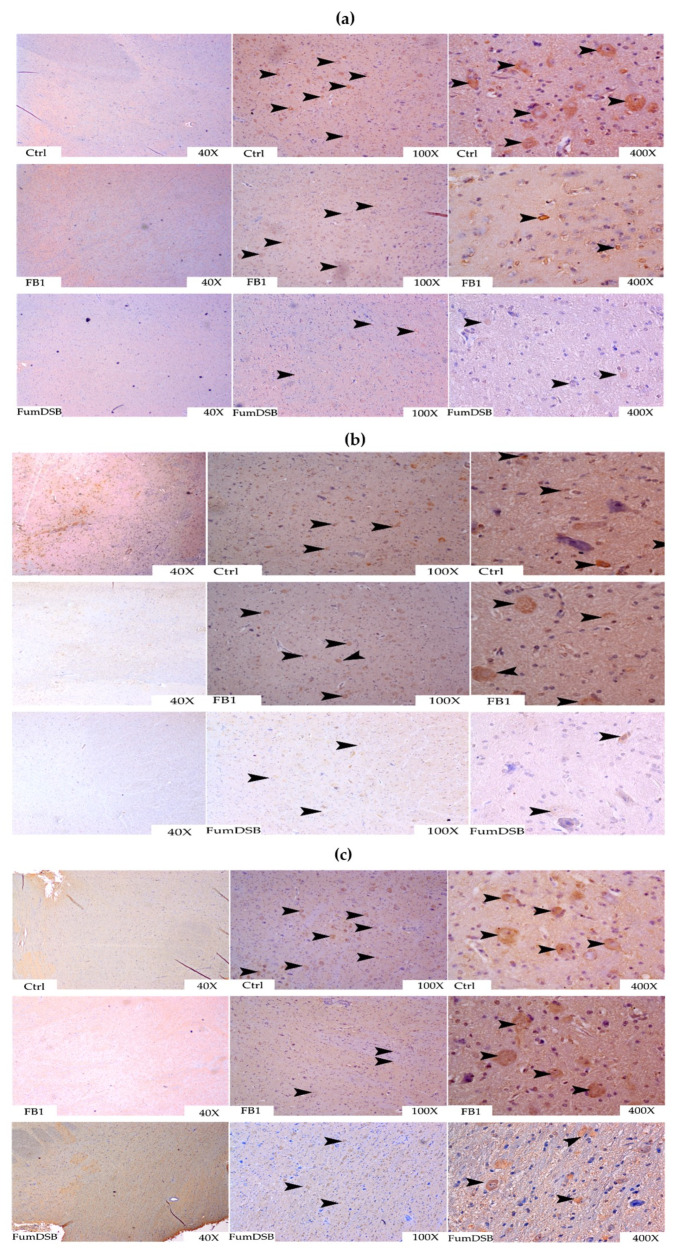
Representative IHC photograph of ghrelin/obestatin/GHSR in the hypothalamus. (**a**–**c**) means the results of ghrelin, obestatin and GHSR, respectively. Ctrl, FB_1_ and FumDSB refer to different treatment. 40×, 100× and 400× represent the magnification. Brown positive reactants are emphasized using black arrows.

**Figure 8 toxins-13-00874-f008:**
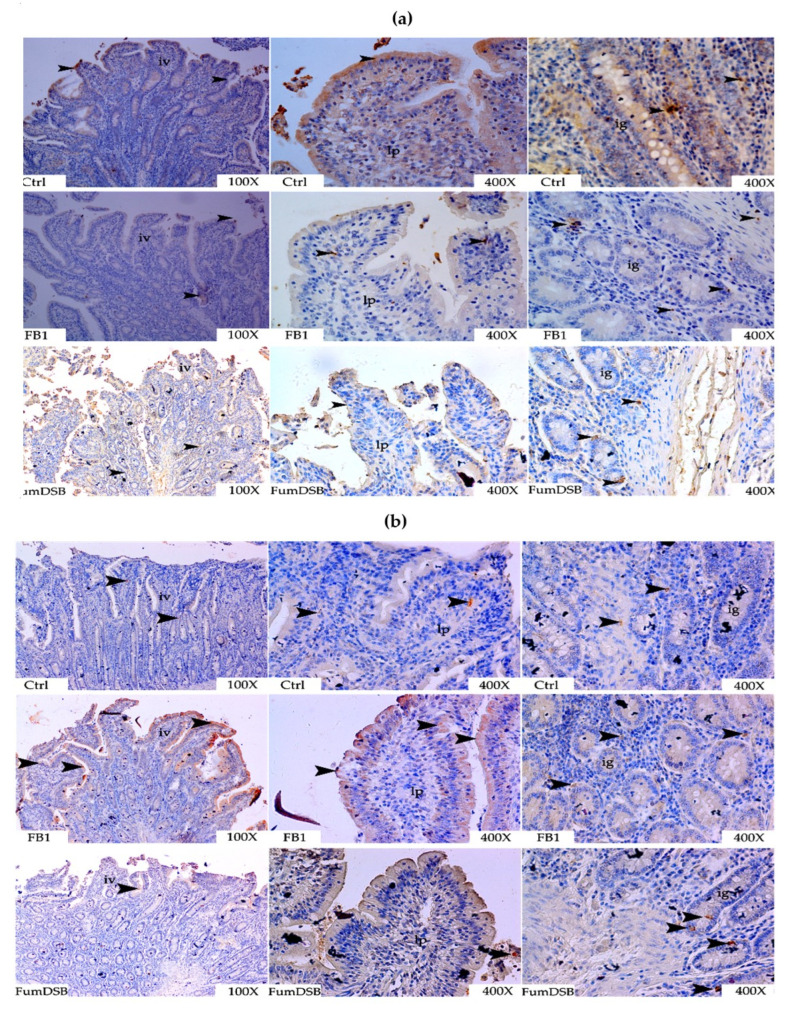
Representative IHC photograph of ghrelin/obestatin/GHSR in the jejunum. (**a**–**c**) means the results of ghrelin, obestatin and GHSR, respectively. Ctrl, FB_1_ and FumDSB refer to different treatment. iv means intestinal villus, ie means intestinal epithelium and ig means intestinal gland. 100× and 400× represent the magnification. Brown positive reactants are emphasized using black arrows.

**Table 1 toxins-13-00874-t001:** Effect of FB_1_ on the production performance of growing pigs.

Group	Initial Weight (kg)	Final Weight (kg)	ADG (kg)	ADFI (kg)	F/G
Control	20.68 ± 1.21	49.75 ± 3.26 ^a^	0.692 ± 0.055 ^a^	1.66 ± 0.15 ^a^	2.39 ± 0.27 ^c^
FB_1_	20.80 ± 1.35	45.81 ± 3.61 ^c^	0.596 ± 0.057 ^c^	1.56 ± 0.23 ^c^	2.62 ± 0.42 ^a^
FumDSB	20.45 ± 0.75	47.53 ± 2.58 ^b^	0.645 ± 0.052 ^b^	1.62 ± 0.21 ^b^	2.50 ± 0.26 ^b^

^a, b, c^ The values with different small letters in the same column differ significantly (*p* < 0.05). Data are expressed as means ± standard deviations, *n* = 8. ADG, average daily body weight gain; ADFI, average daily feed intake; F/G, the ratio of feed intake to body weight gain.

**Table 2 toxins-13-00874-t002:** Composition and nutrient levels of the basal diet (dry matter basis).

**Ingredients (100%)**		**Nutrient Levels (%) ^2^**	
Corn	66.0	Dry matter	87.31
Wheat bran	5.0	Crude protein	17.4
Soybean meal	21.0	Calcium	0.78
Extruded full-fat soybean	4.0	Total phosphorus	0.52
Premix ^1^	4.0	Methionine + Cysteine	0.60
Total	100	Lysine	0.96
		ME (MJ/kg)	12.79
**Content of Mycotoxin (μg/kg, ppb)**			
Fumonisin B_1_ (FB_1_ + FB_2_)	256.0 ± 21.0		
Zearalenone (ZEA)	18.3 ± 2.5		
DON	<LOD **^3^**		
Aflatoxin B_1_ (AFB_1_)	<LOD **^3^**		

^1^ The premix provided the following in the diet (per kg): VA 12, 000 IU; VD_3_ 2, 000 IU; Ca 2.1 g; VE 40 mg; VK_3_ 1.0 mg; VB_1_ 1.0 mg; VB_2_ 3.7 mg; VB_6_ 3 mg; VB_12_ 0.02 mg; niacin 15 mg; folic acid 0.6 mg; D-pantothenic acid 15 mg; choline 250 mg; Mn 40 mg; Fe 100 mg; Zn 100 mg; Cu 180 mg; I 0.30 mg; Se 0.30 mg; Co 1.0 mg. ^2^ ME (metabolizable energy) was a calculated value, while the others were measured values. **^3^** LOD refers to limit of detection, the LOD for DON and AFB_1_ was 10 ppb and 0.5 ppb, respectively.

**Table 3 toxins-13-00874-t003:** Primer sequences of relating genes used for qRT-PCR.

Target Gene	GenBank No.	Primer Sequence (5′-3′)
NPY-F	NM_001256367.1	TCACCAGGCAGAGATACGGA
NPY-R		ACACAGAAGGGTCTTCGAGC
PYY-F	NM_001256528.1	GGAGGAGCTGAGCCGCTACTAC
PYY-R		GCTGTCACGTTTCCCATACCTCTG
5-HT2A-F	NM_214217.1	ATGCAGTCCATCAGCAACGA
5-HT2A-R		ATGACGGCCATGATGTTGGT
GHRL-F	NM_213807.2	GCAGCCAAACTGAAGCCC
GHRL-R		AACTTGATCCCAACATCACAGG
GHSR-F	NM_214180.1	CGGAGTGGAGCATGATAACG
GHSR-R		ACAGGCAGGAAGAAGAAGACA
GPR-39-F	XM_021074555.1	TAGCCGTTGGACTGTGTTCC
GPR-39-R		GGTCACAACGATCAGCCTCA
ACTB-F	XM_021086047.1	GTGCGGGACATCAAGGAGAA
ACTB-R		CGTAGAGGTCCTTGCGGATG
GAPDH-F	NM_001206359.1	TCGGAGTGAACGGATTTGGC
GAPDH-R		TGACAAGCTTCCCGTTCTCC

## Data Availability

Not applicable.

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
