# Peer review of "FumDSB Can Reduce the Toxic Effects of Fumonisin B1 by Regulating Several Brain-Gut Peptides in Both the Hypothalamus and Jejunum of Growing Pigs"

_toxins, 2021, doi:10.3390/toxins13120874_

Round 1

Reviewer 1 Report

The manuscript has been examined. It is similar to another article in the designs and structures ever been submitted to the "Toxins", No.1318508 entitled, “Effects of Fumonisin B1 on inflammatory cytokines and appetite-related neuropeptides both in hypothalamus and jejunum of growing pigs."

Author Response

Thank you very much for your professional and meticulous work. The manuscript "Effects of Fumonisin B1 on inflammatory cytokines and appetite-related neuropeptides both in hypothalamus and jejunum of growing pig (Toxins-1318508)" has been submitted by our lab previously. In that manuscript, we focused on the role of fumonisins on hypothalamus and jejunum of growing pig, and did not include the research content of FumDSB. Finally, due to the lack of rigorous grouping in the entire trial design, it was not accepted by Toxins magazine. In the new submission, we reorganized the new manuscript based on the opinions of experts and our own work.

Reviewer 2 Report

General comments

This study was presented an interesting topic for the readers of Toxins. The paper presents the detoxified effect of the biodegraded enzyme (FumDSB) by investigating growth performances and brain-gut peptides in growing pigs. Although the experimental design appears adequate and the results appear to be interesting, several significant issues emerged during the revision of the paper.

Title

Comment 1: “Degrade” is more suitable for the structure changing of mycotoxins rather than the toxic effect (I also suggest you can switch to “toxicities). The more appropriate word is “detoxify.” According to the above reason, I suggest modifying the title to the following form: “FumDSB Detoxify the Toxicities of Fumonisin B1 by Regulating Several Brain-Gut Peptides in Both of the Hypothalamus and Jejunum of Growing Pigs”. Please also consider revising other corresponding written narratives in the current manuscript.

Abstract

Comment 1 (Line 5): Fumonisins have 4 different types (A, B, C, and P), therefore FB is not the correct abbreviation for Fumonisin. It should be correct to Fumonisin B (FB).

Comment 2 (Line 13-16)

(1) In my opinion, “feed intake” was more suitable than “food intake” for growing pig. Please also consider revised other corresponding written narrative in current manuscript.

(2) It should be “brain-gut”, not be “gut-brain”. The narrative of the whole manuscript should be consistent.

(3) FB1 should be subscript, such as your result shows. The narrative of the whole manuscript should be consistent. Please check the entire article.

(4) Please consider correcting to “…. FB1 exposure reduced feed intake and weight gain of growing pigs and changed the expression and distribution of several appetite-related brain-gut peptides….”.

Key Contribution

Comment 1: Redundant with the title. This section should be more emphasize the importance of this study.

Introduction

Comment 1 (L47): References 4 and 10 did not discuss the chemical detoxifying reagents that lead to secondary pollution and safety hazards. Please change the citation references.

Comment 2 (L57): Please write the suitable pH and temperature range.

Comment 3 (L79-82): To be rewritten. It is difficult to understand what the author wants to express.

Result

Comment 1 (Table 1)

(1) The unit of ADG should change to kg. ADG and ADFI should be the same unit.

(2) Why is the control group not compared with the other two treatment groups?

Comment 2 (Figure 1 a)

The FumDSB’s legend of Figure 1a was covered by Figure 1b

Comment 3

In the result section, the author used a lot of descriptions with “after adding FumDSB”. In my opinion, the meaning of after adding FumDSB is closer to the treatment of the FB1 before giving FumDSB. However, the three treatment groups were carried out simultaneously in this study according to the author’s description. I suggest that the relevant description should be rewritten to avoid misunderstandings. Please also consider revising other corresponding written narratives in Result.

Comment 4

It should be GPR39, it was not GRP39 (L176, 178, 182, 314, and 324).

Comment 5

The title of Figure 7 was ghrelin/obestatin signals, it should be GHRL description. Why was it described with GHSR in Figures 7b and 7c (L219 and 221)?

Discussion

Comment 1 (L287-290): These sentences need to add citation.

Comment 2:

(1) Information about the detailed mechanism of FB1 inducing 5-HT secretion and then inhibiting the appetite should be mentioned.

(2) The discussions of growth performance and peptides expression were not connected well. Please add more related references and discussion.

Material and Methods

Comment 1:

Many chemicals, instruments, and materials didn’t provide detailed information such as serial numbers and manufacturer information.

Examples are as follows,

(1) L343: Missing the country information of university.

(2) Missing the manufacture information and series number of paraformaldehyde (L379), PVDF membranes (L411), DAB (L427), and microscope (L429).

Please check the entire part of Methods and Materials

Comment 2:

There must be a space between the number and the unit. For example 1 g (L348) and 1 L(L349).

Comment 3 (L351):

The correct concentration should be “5” mg/kg, it was not “10” mg/kg.

Comment 4 (Table 2)

(1) The results were shown on the dry matter (DM) basis but no DM (%) was shown on this Table.

(2)The unit of mycotoxin concentration should be μg/kg (ppb).

Comment 5 (Line 426):

(Wang et al., 2020) is not the correct citation format for Toxins, as well as it is not shown in References.

References

Comment 1: The year of reference #11 was repeated twice.

Author Response

Dear Journal Reviewer:

Great thanks for your helpful and professional suggestions and the comments on our manuscript “FumDSB Detoxify the Toxicities of Fumonisin B1 by Regulating Several Brain-Gut Peptides in Both the Hypothalamus and Jejunum of Growing Pigs” (Toxins-1468891). Those comments are all valuable and very helpful for revising and improving our paper, as well as the important guiding significance to our researches. We have studied comments carefully and replied all of the points raised by referees, and the revised portion are marked using using the “Track Changes” function in the manuscript. The main corrections and explain in the revised manuscript are listed in attachment.

Reviewer 3 Report

This manuscript has evaluated the detoxification effects of FumDSB on Fumonisin B1-induced negative effects in the growing pigs. This is an interesting topic. The experimental was well-designed and the data was well-presented. The writing need to be improved. The following revision could improve the quality of the paper.

Major comments:

  1. What is the structure of the biodegradated metabolites of Fumonisin by FumDSB?
  2. Since the study was focus on the enzyme activity of FumDSB, it is important to check the biodegradate metabolite of Fumonisin? Or measuring the residue of Fumonisin in the tissues or serum. It is not that important to detect the molecular biomarkers.

Minor comments:

  1. Title, “FumDSB Can Degrade the Toxic Effects…”. The sentence is not right and could be correct to “FumDSB Can Reduce the Toxic Effects…”.
  2. It is better to add how many changes happened in the abstract for the indexes.
  3. Line 91, please writing the information of “(M±SD, n=8)” and others in the footnote of the table. Please check all the information.
  4. Line 92, please correct “(p < 0.05)” to “(p < 0.05)”. All the P value should be written in italic.
  5. Lines 95-96, please use the different letter to substitute the “*” for the significant changes. Please correct this throughout the paper.
  6. Figure 2, why the different housekeeping protein has been used in the same tissue?
  7. Lines 131-143, please make the consist to use “Figure” and “Fig” throughout the paper.

Author Response

(The authors gave the same response as above.)

Round 2

Reviewer 1 Report

The revised manuscript toxins-1468891 entitled, “FumDSB detoxify the toxicities of Fumonisin B1 by regulating several brain-gut peptides in both the hypothalamus and jejunum of growing pigs” has been reviewed. The expert opinions from another two respectable reviewers were also read carefully and taken as reference on decision.

In the previous manuscript No. 1318508 from the same group, the effects of fumonisin B1 on the appetite-related neuropeptides and inflammatory cytokines including MC4R, AgRP, POMC, INF-γ, NF/κB/STAT3 signaling, IL-2, -6, -8, and TNF-α both in hypothalamus and jejunum of 16 growing pigs was studied. In current manuscript, the effects of fumonisin B1 on the brain-gut peptides (NPY, PYY, 5-HT2A) and ghrelin/obestatin/GHSR (GHRL, GHSR, GPR-39) both in hypothalamus and jejunum of 16 growing pigs was studied. The detoxifying effects of FumDSB were studied in another 8 growing pigs. They have similar study design and methodology.

The concerns remained. The manuscript was not adequately prepared on submission including the references, as mentioned by another reviewer. The resolution of the figures was poor and difficult to read. The authors did not describe and discuss the mechanisms of fumonisin B1 and FumDSB globally involved the abovementioned molecules, which may contribute greatly to the understanding of mycotoxins.

Author Response

Dear Reviewer:

Great thanks for your helpful comments on our manuscript. The revised portion are marked using the “Track Changes” function in the manuscript. And the responds to the reviewer’s comments are as following.

  1. In the previous manuscript No. 1318508 from the same group, the effects of fumonisin B1 on the appetite-related neuropeptides and inflammatory cytokines including MC4R, AgRP, POMC, INF-γ, NF/κB/STAT3 signaling, IL-2, -6, -8, and TNF-α both in hypothalamus and jejunum of 16 growing pigs was studied. In current manuscript, the effects of fumonisin B1 on the brain-gut peptides (NPY, PYY, 5-HT2A) and ghrelin/obestatin/GHSR (GHRL, GHSR, GPR-39) both in hypothalamus and jejunum of 16 growing pigs was studied. The detoxifying effects of FumDSB were studied in another 8 growing pigs. They have similar study design and methodology.

---  Great thanks for your serious and rigorous work. The previous manuscript Toxins-1318508 was also completed and submitted by our laboratory, and the animals used are the same batch of pigs as the experiment in this manuscript. Methods of western blot, qRT-PCR and immunohistochemistry were used to examine the inflammatory factors and brain-gut peptides in the hypothalamus and jejunum from the perspective of brain-gut regulation. Because of the patent application and other factors, in previous manuscript Toxins-1318508, animal group was only divided into the control group and the FB1 group (additional amount at 5mg/kg), and lack of the addition of FumDSB,  which focused on analyze the toxic effect of FB1 on the hypothalamus and jejunum. But unfortunately it was rejected because of the grouping problem. At present, we have obtained the permission of the FumDSB R&D staff, and the results of carboxylesterase FumDSB can be published publicly. Therefore, in the newly submitted manuscript, we focused on the effect of FumDSB. 

  1. The concerns remained. The manuscript was not adequately prepared on submission including the references, as mentioned by another reviewer. The resolution of the figures was poor and difficult to read. The authors did not describe and discuss the mechanisms of fumonisin B1 and FumDSB globally involved the abovementioned molecules, which may contribute greatly to the understanding of mycotoxins.

--- Great thanks for your comments. In the revised manuscript, we checked the references again and adjusted the resolution ratio of the graphics. Preliminary studies have proved that FumDSB1 can degrade FB into non-toxic HFB1 and has relatively stable properties, which is suitable for animal use. And detailed results and analysis were reported by Li et al.( 2021). However, the previous studies was carried out under laboratory conditions, and it is not clear whether effective and safe after FumDSB being in taken by animals. The main purpose of this manuscript was to examine the effect and safety of FumDSB as a detoxification enzyme preparation from the perspective of brain-gut regulation. The results showed that FumDSB1 has good effects and safety, and has the value of in-depth research. However, due to the first in-vivo experiment in animals, its mechanism of FumDSB is mainly achieved on the basis of degradation of FB1, but its metabolism and other effects in the body need to be further studied.

Li, Z.; Wang,Y.; Liu Z.; Jin S.; Pan, K.; Liu, H.;Liu,T.; Li X.; Zhang C.; Luo, X.; , Song Y.; Zhao J.; Zhang,T. Biological detoxification of fumonisin by a novel carboxylesterase from Sphingomonadales bacterium and its biochemical characterization. Int J Biol Macromol. 2021, 169, 18–27.

Sincerely

Reviewer 2 Report

My comments have been appropriately answered and revised, and I don’t have any other comments. This manuscript could be accepted in the present form.

Author Response

Great thanks for your approval

Reviewer 3 Report

Thanks for you clarification. No further comments.

Author Response

Great thank for your approval

This manuscript is a resubmission of an earlier submission. The following is a list of the peer review reports and author responses from that submission.

Round 1

Reviewer 1 Report

Manuscript toxins-1318508 entitled, “Effects of Fumonisin B1 on inflammatory cytokines and appetite-related neuropeptides both in hypothalamus and jejunum of growing pigs.” submitted to the section “Mycotoxins” was reviewed.

The authors studied the effects of fomonisin B1 through clinical observations, histopathology, qRT-PCR, and immunohistochemistry and illustrated the neuro-inflammatory responses between hypothalamus and jejunum. The Table 1 was briefly constructed. Representative microscopic photographs were shown. Histogram was used to show the differences in mRNA expression

The following comments were provided. 

  1. The demonstration of each portion of the effects is adequate and reasonable. But, the correlation and the causal relationship need be studied. The proposed mechanism was not clearly mentioned. In the line 15 or line 274, “…interfere the expression of NF/κB/STAT3 signaling” seems not a good term. 
  2. The results did not add more on current knowledge about the fumonisin. As the authors mentioned, the effects of fumonisin on the pro-inflammatory cytokines like they studied and anti-inflammatory cytokines like IL-10 is controversial. More discussion is required in line 248-272.
  3. The magnification power was not consistent in Figure 6, Figure 7, and the legends. The resolution need be improved to delineate the cellular location of each cytokine, such as the line 148 “…distributed in and around the cytoplasm of …”. 
  4. There are several uncomfortable grammars, such as line 113-115, line 120-122, line 153-162, 177-179, line 212-215.

Author Response

Dear Journal Manager:

Great thanks for your helpful and professional suggestions and the comments on our manuscript “Effects of Fumonisin B1 on inflammatory cytokines and appetite-related neuropeptides both in hypothalamus and jejunum of growing pigs” (Toxins-1318508). Those comments are all valuable and very helpful for revising and improving our paper, as well as the important guiding significance to our researches. We have studied comments carefully and have made corrections which we hope meet with approval. The revised portion are marked using red color in the manuscript. The main corrections and explain in the paper and the responds to the reviewer’s comments are as following:

  1. The demonstration of each portion of the effects is adequate and reasonable. But, the correlation and the causal relationship need be studied. The proposed mechanism was not clearly mentioned. In the line 15 or line 274, “…interfere the expression of NF/κB/STAT3 signaling” seems not a good term. 

-- Great thanks for your advice. In the line 15 or line 274, “…interfere the expression of NF/κB/STAT3 signaling” has been revised to “…that the consumption of FB1 can affect the expression of the NF-κB/STAT3 signaling molecules” in revised manuscript. Meanwhile, the correlation and the causal relationship has been added in discuss.

  1. The results did not add more on current knowledge about the fumonisin. As the authors mentioned, the effects of fumonisin on the pro-inflammatory cytokines like they studied and anti-inflammatory cytokines like IL-10 is controversial. More discussion is required in line 248-272.

-- Great thanks for your advice. More discussion were added in revised manuscript.

  1. The magnification power was not consistent in Figure 6, Figure 7, and the legends. The resolution need be improved to delineate the cellular location of each cytokine, such as the line 148 “…distributed in and around the cytoplasm of …”. 

-- Great thanks for your advice. The magnification power in Figure 6 and Figure 7 in legends have been corrected , as well as the delineation of cytokine has been added in revised manuscript.

  1. There are several uncomfortable grammars, such as line 113-115, line 120-122, line 153-162, 177-179, line 212-215.

-- I apologize for my English expression. In view of our lack of English expression, the grammar and vocabulary of the full text have been revised by professional English editors of MDPI. 

Reviewer 2 Report

  1. The study design of the current research is not sufficient and somewhat unreasonable. At first, it's very crucial to do in vitro study for supporting in vivo study.
  2. Authors must be considered as 'long- and short-term toxin trial' otherwise it will make confusion and queries to the scientific community.
  3. Authors should perform both ELISA and qRT-PCR.

Author Response

Dear Journal Manager:

Great thanks for reviewers’ helpful and professional suggestions and the comments on our manuscript “Effects of Fumonisin B1 on inflammatory cytokines and appetite-related neuropeptides both in hypothalamus and jejunum of growing pigs” (Toxins-1318508). Those comments are all valuable and very helpful for revising and improving our paper, as well as the important guiding significance to our researches. We have studied comments carefully and have made corrections which we hope meet with approval. The revised portion are marked using red color in the manuscript. The main corrections and explain in the paper and the responds to the reviewer’s comments are as following:

  1. The study design of the current research is not sufficient and somewhat unreasonable. At first, it's very crucial to do in vitro study for supporting in vivo study.

-- Great thanks for your valuable comments. In vitro experiments are very important to support in vivo experiments. We used to do research on animal applications, focusing on the effects of animals in vivo. Our next step will be to carry out in vitro tests of toxins and the results of in vivo studies.

  1. Authors must be considered as 'long- and short-term toxin trial' otherwise it will make confusion and queries to the scientific community.

-- Great thanks for your professional advice. This study is a short-term toxin trail, and we will add an explanation in the revised manuscript.

  1. Authors should perform both ELISA and qRT-PCR.

-- Great thanks for your professional advice. On this issue, our explanation is as follows:

In preliminary experiment, we found that the concentration of inflammatory factors in the hypothalamus was rather low, and the concentration of inflammatory factors were measured by general ELISA were not accurate. On the other hand, the sampling volume of hypothalamus is lower, therefore, the development trend of inflammatory factors in limited hypothalamus samples were determined through qRT-PCR and histochemistry methods conjointly in this study. Meanwhile, in some related references, only qRT-PCR was used to detect factors related to feeding in the hypothalamus. Such as:

Tominaga, et al., 2016. Anorexic action of deoxynivalenol in hypothalamus and intestine. Toxicon. 118, 54-60. https://doi.org/10.1016/j.toxicon.2016.04.036.

Fan, et al.,2021.The role of infammatory cytokines in anemia and gastrointestinal mucosal injury induced by foot electric stimulation. Scientifc Reports. 11, 3103. https://doi.org/10.1038/s41598-021-82604-7.

Based on the above reasons, we did not use ELISA to determine the concentration of inflammatory factors and appetite-related neuropeptides.

Round 2

Reviewer 1 Report

The revised manuscript toxins-1318508 entitled, “Effects of Fumonisin B1 on inflammatory cytokines and appetite-related neuropeptides both in hypothalamus and jejunum of growing pigs.” submitted to the section “Mycotoxins” was reviewed.

There was significant improvement after the revision and medical editing. The issue about the varying results of fumonisin on the IL-4 and IL-10 expression. remained not be addressed. The authors skipped the original reference 45, although they mentioned, “These results indicate that the regulation effect of cytokines by FB1 may be associated with different strains, sexes, and doses. Therefore, more data under different conditions must be obtained in future research.” In line 340-342. Another reference 47 was added, in which the IL-4 expression was reduced after fumonisin.

The resolution of Figure 6 and Figure 7 is another concern.

Author Response

Dear Reviewer:

Great thanks for your and reviewers’ helpful and professional suggestions and the comments on our manuscript “Effects of Fumonisin B1 on inflammatory cytokines and appetite-related neuropeptides both in hypothalamus and jejunum of growing pigs” (Toxins-1318508). The revised portion are marked using red color in the manuscript. The main corrections and explain in the paper and the responds to the reviewer’s comments are as following:

Respond to Reviewer 1:

There was significant improvement after the revision and medical editing. The issue about the varying results of fumonisin on the IL-4 and IL-10 expression. remained not be addressed. The authors skipped the original reference 45, although they mentioned, “These results indicate that the regulation effect of cytokines by FB1 may be associated with different strains, sexes, and doses. Therefore, more data under different conditions must be obtained in future research.” In line 340-342. Another reference 47 was added, in which the IL-4 expression was reduced after fumonisin.

--  Great thanks for for your comments and opinions. We revisited part of the literature and discussed the differences in inflammatory factors. The corrected part in revised manuscript has been marked by red.

The resolution of Figure 6 and Figure 7 is another concern. 

-- Great thanks for your suggestion. The resolution of Figure 6 and Figure 7 has been improved in revised manuscript.

Reviewer 2 Report

According to previous studies, the effects of dietary fumonisin B1 (FB1) on regional brain is dose and time-dependent. As CNS is the main target of fumonisins that can cause nerve cell damage. So, time treatment of FB1 (short and long) is very crucial. I wonder how authors select a single dose and time point for the hypothalamus and jejunum? At least a couple of dosages (5mg/kg and 10mg/kg) must be needed to verify the exact mechanism. As a result, study design should be changed.

Author Response

Dear Reviewer:

Great thanks for your and reviewers’ helpful and professional suggestions and the comments on our manuscript “Effects of Fumonisin B1 on inflammatory cytokines and appetite-related neuropeptides both in hypothalamus and jejunum of growing pigs” (Toxins-1318508). The revised portion are marked using red color in the manuscript. The main corrections and explain in the paper and the responds to the reviewer’s comments are as following:

Respond to Reviewer 2

Thank you very much for your professional comments and opinions. For your questions, our response is as follows:

FB1 has very high pollution levels in corn and other grains, but high-dose FB1 is relatively rare in current feed production, and instances of acute poisoning induced by FB1 at high doses are very rare. The results from many references and pre-experiment in our lab proved that a significant loss of appetite has not observed after treating the pigs with FB1 at lower dose. The purpose in this study was to analyze the correlation with appetite suppression of growing pigs induce by FB1 and inflammatory cytokines and appetite-related factors. Therefore, the amount of FB1 added to induce appetite suppression in growing pigs was a basic indicator for us to choose the test dose.

 In many countries, the limit of FB1 in the compound feeds for pigs is 5mg/kg. FB1 is a potential health risk for growing pigs, as dietary exposure to approximately 5.0 mg/kg FB1 or higher for 6 months can cause neurochemical changes in the brain and lead to adverse physiological reactions. And this amount showed inhibition of feeding in our pre-experiment, and there were no obvious signs of clinical toxicity, though affecting the changes in cytokine expression. Therefore, as a short-term toxin test in this study, the amount of 5.0 mg/kg FB1 was selected to analyze the correlation with appetite suppression induce by FB1 and inflammatory cytokines and appetite-related factors.

In addition, the discussion about the different effects of FB1 on inflammatory factors has been re-corrected in revised manuscript, which marked using red color.

Round 3

Reviewer 2 Report

I have rejected this manuscript "Effects of Fumonisin B1 on inflammatory cytokines and appetite-related neuropeptides both in hypothalamus and jejunum of growing pigs" a couple of times and I think it's not a quality research article to publish in the assigned journal.